# Insecticidal Effects of Transgenic Maize Bt-Cry1Ab, Bt-Vip3Aa, and Bt-Cry1Ab+Vip3Aa against the Oriental Armyworm, *Mythimna separata* (Walker) in Southwest China

**DOI:** 10.3390/toxins16030134

**Published:** 2024-03-04

**Authors:** Zhenghao Zhang, Xianming Yang, Wenhui Wang, Kongming Wu

**Affiliations:** 1State Key Laboratory of Ecological Pest Control for Fujian and Taiwan Crops, Fujian Agriculture and Forestry University, Fuzhou 350002, China; zhenghaozzh@163.com; 2State Key Laboratory for Biology of Plant Diseases and Insect Pests, Institute of Plant Protection, Chinese Academy of Agricultural Sciences, Beijing 100193, China; yangxianming@caas.cn (X.Y.); w975480209@163.com (W.W.); 3Institute of Insect Sciences, College of Agriculture and Biotechnology, Zhejiang University, Hangzhou 310058, China

**Keywords:** *Mythimna separata*, Bt maize, pest management, population dynamics, China

## Abstract

The oriental armyworm, *Mythimna separata* (Walker), an important migratory pest of maize and wheat, is posing a severe threat to maize production in Asian countries. As source areas of spring–summer emigratory populations, the control of *M. separata* in southwestern China is of great significance for East Asian maize production. To assess the toxicity of Bt maize against the pest, bioassays of Bt-(Cry1Ab+Vip3Aa) maize (event DBN3601T), Bt-Cry1Ab maize (event DBN9936), and Bt-Vip3Aa maize (event DBN9501) were conducted in Yunnan province of southwest China. There were significant differences in insecticidal activity between the three Bt maize events, and DBN3601T presented the highest insecticidal role. The results also indicated that the insecticidal effect of various Bt maize tissues took an order in leaf > kernel > silk, which is highly consistent with the expression amounts of Bt insecticidal protein in leaf (69.69 ± 1.18 μg/g), kernel (11.69 ± 0.75 μg/g), and silk (7.32 ± 0.31 μg/g). In field trials, all larval population densities, plant damage rates, and leaf damage levels of DBN3601T maize were significantly lower than the conventional maize. This research indicated that the DBN3601T event had a high control efficiency against *M. separata* and could be deployed in southwest China for the management of *M. separata*.

## 1. Introduction

The oriental armyworm, *Mythimna separata* (Walker) (Lepidoptera: Noctuidae), is a serious cereal pest that mainly damages maize, wheat, rice, sorghum, and other gramineous crops, and also damages fruit trees, oilseeds, vegetables, and other plants [1]. Oriental armyworm can migrate over long distances and is widely distributed among several countries and regions in Asia and Australia [2]. In China, it has been found in all regions except Tibet [1]. For maize, this pest mainly feeds on leaves and infests silks and young kernels [3].

Southwest China is the main maize-producing area in China [4], mainly including the Sichuan, Chongqing, Guizhou, and Yunnan provinces. Yunnan province is affected by southwesterly winds in the summer and northeasterly winds in the winter, forming the migration pattern for migratory pests that migrate northward in the spring and summer, and return in the fall and winter, and it is a migration corridor for major migratory pests such as the *Spodoptera frugiperda* (J.E. Smith), *M. separata*, and *Spodoptera litura* [5,6]. The southern part of southwest China is the year-round breeding area for the oriental armyworm [7]. Hence, the control of oriental armyworm in this region can avoid population outbreaks in China, especially reduce the pest pressure in the main maize-producing areas, e.g., the Huang-Huai-Hai plain and northeast China.

For a long time, chemical control has been the main means in controlling the oriental armyworm. However, the long-term use of pesticides has seriously polluted the ecological environment, adversely affected non-target organisms, and caused pesticide resistance among pests [8]. In 2021, the oriental armyworm was found to be 126-fold more resistant to cypermethrin in the Punjab region of Pakistan [9]. In China, this pest has evolved resistance to the most used insecticides, including pyrethroids, organophosphates, diamides, and abamectin [10]. Zhao et al. (2017) have found that some field populations in China have developed resistance to chlorpyrifos [11]. The pesticide resistance of oriental armyworms reduced insecticidal efficiency, therefore there is an urgent need for a green and efficient control method.

The commercial cultivation of transgenic crops expressing *Bacillus thuringiensis* (Bt) toxins provides new ways to control pests [12]. Bt crops can effectively control target pests, avoid the problem of pesticide residues, reduce the cost of pest control, and protect farmland biodiversity [13]. Bt maize which is genetically modified by inserting exogenous gene fragments expressing insecticidal proteins into common maize genes is commonly used to control lepidopteran and coleopteran pests. The commonly used Bt proteins are the Cry and Vip classes. Cry includes Cry1Ab, Cry1Ac, Cry1C, Cry1F, Cry1A.105, etc. [14]. Vip includes Vip3Aa19 and Vip3Aa20, etc. [15]. In 2020–2021, Bt-Vip3Aa19 maize (event DBN9501), Bt-Cry1Ab maize (event DBN9936), and Bt-(Cry1Ab+Vip3Aa19) maize (event DBN3601T) were issued the transgenic biosafety certificate by the Chinese government. DBN3601T maize resulting from the stacking of DBN9936 and DBN9501 has enhanced the control efficiency to a broader spectrum of target pest, whereby the content of Bt toxin (Cry1Ab+Vip3Aa) in DBN3601T maize leaves was 81.29 ± 1.20 μg/g, the Cry1Ab was 76.54 ± 0.60 μg/g in DBN9936 maize leaves, and Vip3Aa in DBN9501 maize leaves was 5.08 ± 0.08 μg/g [16]. Some studies have shown that the expression of insect-resistant proteins in these transformation events has reached high dosage levels against fall armyworm and can effectively control the pest [17].

There have been many studies on the control efficiency of Bt maize to fall armyworm [18], but few on oriental armyworm. Our study aimed to assess the control efficiency of the leaf, kernel, and silk tissues of Bt-(Cry1Ab+Vip3Aa) (event DBN3601T), Bt-Cry1Ab (event DBN9936), and Bt-Vip3Aa (event DBN9501) maize to this first, third, and fifth stage pest using the tissue bioassay method, and also aimed to investigate the field control efficiency of DBN3601T maize to this pest using the mesh cage method. At the same time, the natural occurrence and damage of this pest in DBN3601T maize was also evaluated in Puer, Yunnan province, so as to provide a scientific basis for the control of this pest by using Bt maize.

## 2. Results

### 2.1. Comparison of Insecticidal Effects of Three Bt Maize Events against the Oriental Armyworm

The Bt protein amount in DBN3601T varied significantly across tissues (F_2,6_ = 5898.47, *p* < 0.001), with much higher levels in both V5 leaf (69.69 ± 1.18 μg/g) and R2 kernel (11.69 ± 0.75 μg/g), exceeding that of R1 silk tissue (7.32 ± 0.31 μg/g) (Table 1). This variation in protein content aligns with the varying lethality observed across these tissues against the pest, in the order of leaf > kernel > silk (Figure 1B).

The results showed that Bt maize event, tissue, and stage had statistically significant effects on corrected mortality, respectively (event, *χ*^2^ = 657.70, *p* < 0.001; tissue, *χ*^2^ = 315.38, *p* < 0.001; stage, *χ*^2^ = 687.51, *p* < 0.001). After 4 days, where the larvae infested the DBN3601T event, leaf tissue exhibited higher corrected mortality than the other events or tissues (Figure 1A,B). The first-stage larvae were more susceptible to Bt maize than the third- and fifth-stage larvae, but no significant difference was observed between third-stage and fifth-stage larvae (*p* = 0.131) (Figure 1C). When first-stage larvae infested leaf tissues for 4 days, the corrected larvae mortality was 98.00 ± 0.58% on DBN3601T, 80.00 ± 2.89% on DBN9936, and 31.94 ± 9.10% on DBN9501, respectively; when first-stage larvae infested kernel tissues for 4 days, the corrected larvae mortality was 57.87 ± 6.12% on DBN3601T, 50.93 ± 4.90% on DBN9936 and 11.09 ± 3.66% on DBN9501, respectively; while the corrected mortality of the first-stage larvae infesting silk on DBN3601T, DBN9936, and DBN9501 on 4 days were only 55.09 ± 7.28%, 50.00 ± 1.16%, and 5.52 ± 3.50%, respectively (Figure 1D).

The corrected mortality line graphs of the oriental armyworm infesting different tissues from DBN3601T were quite different. Both tissue (*χ*^2^ = 812.20, *p* < 0.001) and stage (*χ*^2^ = 153.32, *p* < 0.001) significantly affected survival days. The corrected mortality of the first-stage larvae infesting leaves increased the fastest, and the corrected mortality was 10.00 ± 2.89% at 2 days of infestation and 95.00 ± 1.53% at 3 days of infestation; the corrected mortality of different-stage larvae feeding on DBN3601T maize leaf tissues reached 100% on days 6 (first stage), 9 (third stage), and 10 (fifth stage), respectively (Figure 2A). The corrected mortality of first-, third-, and fifth-stage larvae feeding on DBN3601T maize kernel reached 100% on days 14, 14, and 13, respectively (Figure 2B). Some first- (68.98 ± 0.93%), third-stage larvae (23.15 ± 0.93%), and fifth-stage (30.56 ± 0.80%) larvae feeding on silk tissue were poisoned to death on day 14 (Figure 2C). The corrected mortality curves exhibited similar trends for both third-stage larvae and fifth-stage larvae feeding on the kernel, as well as for both third stage larvae and fifth-stage larvae feeding on the silk (Figure 2B,C). After feeding on the leaf, the survival days of first-stage larvae (1.77 ± 0.45) were significantly lower than third- (4.41 ± 0.18) and fifth-stage larvae (4.33 ± 0.17), and there were no significant differences between third-stage larvae and fifth-stage larvae (*p* = 0.943); after feeding on the kernel, the survival days of first-stage larvae (3.51 ± 0.21) were significantly lower than the third- (5.27 ± 0.20) and fifth-stage larvae (5.13 ± 0.16), and there were no significant differences between third-stage larvae and fifth-stage larvae (*p* = 0.829); after feeding on the silk, the survival days of third-stage larvae (21.29 ± 0.73) were significantly higher than first- (10.86 ± 1.01) and fifth-stage larvae (11.54 ± 1.67) in the larval phase and there were no significant differences between first-stage larvae and fifth-stage larvae (*p* = 0.397) in the larval phase (Figure 2D).

### 2.2. Field Control Efficacy of DBN3601T Maize on the Oriental Armyworm Larvae in a Mesh Cage

When the first-stage larvae infested V5 maize plants, the number of larvae per 100 plants (Z = −3.22, *p* = 0.001), leaf damage score (Z = −4.73, *p* < 0.001), and plant damage incidence (Z = −4.79, *p* < 0.001) of the DBN3601T maize were significantly lower than those of conventional maize plants after 4 to 20 days infestation, respectively (Figure 3). On the DBN3601T maize, the highest number of larvae per 100 plants was 102.7 ± 2.7 at day 4 after infecting, and the lowest was 0 on days 12 to 20; in conventional maize, the highest was 123.7 ± 2.2 at day 4 and the lowest was 60.0 ± 7.2 at day 20 (Figure 3A). The leaf damage score, in conventional maize, was 1.99 ± 0.10 (day 4) to 6.46 ± 0.23 (day 20), higher than that in DBN360T maize at 1.29 ± 0.15 (day 4) to 0 (day 20) (Figure 3B). The plant damage incidence of the DBN3601T maize was 22.00 ± 11.01% on day 4 and 0% on day 20, but conventional maize was 70.67 ± 7.42% on day 4 and 81.33 ± 12.88% on day 20 (Figure 3C). The control efficacy of DBN3601T was quite different among the days of pest infestation (one-way ANOVA, F = 212.53, *p* < 0.001), e.g., 24.65 ± 4.54% at 4 days after infestation, but can reach 94.05 ± 2.28%, 100%, 100%, and 100%, on 8-, 12-, 16-, and 20-days post infestation, respectively (Figure 3A).

### 2.3. Naturally Occurring Population Dynamics of Oriental Armyworm in the Bt-(Cry1Ab+Vip3Aa) (Event DBN3601T) Maize Field

For the naturally occurring populations in the field in 2023, two oriental armyworm generations occurred during the whole development maize stage in Puer, Yunnan province. On 7 July, the first generation peaked at maize growth stage V7, with 36.0 ± 2.0 larvae per 100 plants in conventional maize, but only 2.7 ± 0.3 larvae per 100 plants in DBN3601T maize. On 11 August, during the VT growth stage, the second generation peaked at 5.0 ± 0.6 larvae per 100 plants in conventional maize, but 0 larvae occurred in the DBN3601T maize field. There was a significant difference in the control efficacy of the DBN3601T maize at different growth stages (H = 28.66, *p* = 0.001). At stages V6 and V7 with a high larvae abundance (>30 larvae per 100 plants), the control efficacy was 93.99 ± 1.59% and 91.94 ± 1.94%, and it was 100.00% during the other growth stages with a lower larval abundance (<10 larvae per 100 plants) (Figure 4).

## 3. Discussion

In our study, the bioassay shows that the DBN3601T event maize, expressing Cry1Ab and Vip3Aa proteins had better control efficacy against oriental armyworm compared to the DBN9936 maize event expressing Cry1Ab and the DBN9501 maize event expressing Vip3Aa. Previous studies have shown that Syngenta Bt176 (Cry1Ab) and Bt11 (Cry1Ab) maize, which were first commercially planted in 1996 in the United States and other countries, and the Monsanto MON810 (Cry1Ab) maize event showed significant resistance to *Ostrinia nubilalis* (Hübner) [19], similar to our results that DBN9936 could control oriental armyworm more than 80%. In contrast to our results, the MIR162 (Vip3Aa20) maize event can efficiently control *S. frugiperda*, *Helicoverpa zea*, and *Striacosta albicosta* (Smith) [20]. Other studies had shown that the Bt11 × MIR162 event maize, which was genetically stacked Cry1Ab and Vip3Aa, similar to DBN3601T, planted in Brazil in 2014, performed better than the single-gene Bt11 and the MIR162 event maize against *H. zea* [21], which was the same as the results of our study. Furthermore, we observed significant differences in insect resistance among the different tissues of DBN3601T, DBN9936, and DBN9501 maize. Bt toxin change across the different tissues of DBN3601T maize is due to various levels of protein expression, and the leaf exhibited the highest Bt protein concentration (69.69 ± 1.18 μg/g), followed by that in kernel tissue (11.69 ± 0.75 μg/g) and silk tissue (7.32 ± 0.31 μg/g). This pattern mirrors the tissue-specific expression observed in MIR162, MON810, and MON88017 (Cry3Bb1) maize varieties [22,23,24]. In our study, we also observed that the older oriental armyworm larvae were less susceptible to DBN3601T maize. However, in our field trials, DBN3601T maize control efficacy nearly reached 100%, with a lower larval population abundance, leaf damage score, and plant damage incidence than that of conventional maize.

Chemical control is the primary method in the traditional management of oriental armyworms, but the prolonged use of chemical pesticides has led to varying degrees of pesticide resistance in these populations [25]. The emergence of Bt maize can effectively alleviate this problem [26]. The effect of Bt maize on target pests is mainly related to the type of insect-resistant protein it expresses. The relevant studies indicate that the Cry1Ab protein is highly effective at killing oriental armyworms, while the Vip3Aa protein can poison fall armyworm, but cannot poison oriental armyworm with hih efficiency [27]. Thus, when we choose different Bt maize events to control pests, we should ensure that they are highly efficient and can broadly control the target pests. Since the invasion of fall armyworm in China in 2018, they have seriously threatened China’s agricultural production [28]. However, planting DBN3601T event maize in Yunnan can efficiently limit the damage of fall armyworms to maize [29], and our study confirmed that DBN3601T maize also had a good control effect on oriental armyworms. In addition, the adult moths of oriental armyworms prefer to lay their eggs in dried leaf sheaths and withered grass [30], and during a large-scale outbreak, the larvae will crawl and migrate to other host plants nearby after eating all the host plants, which can lead to Bt maize fields being exposed to feeding by late-stage larvae. Exploring the resistance of Bt maize to older larvae can be used to better evaluate its control effect. Previous studies have shown that the maize leaves expressing Cry1Ab protein are 100% lethal to the oriental armyworm at the bulimic feeding stage [31], which is consistent with our results, but our study found that the DBN3601T event maize silk tissues did not reach 100% lethality rate against older oriental armyworm larvae. Therefore, in the practical application of DBN3601T event maize, maize growers should ensure the timely removal of weeds from the field, as well as the timely monitoring of pest population dynamics in the field, especially during the maize reproductive stages. If necessary, spraying pesticides and using other methods to carry out integrated prevention and control. The protein expression levels in Bt crops are influenced by internal regulatory factors, as well as external factors such as environmental conditions (temperature, altitude, etc.) and human factors (fertilization levels, weed management, etc.) [32,33,34]. Therefore, the field management and planting environment of Bt maize need to ensure the normal growth and development of maize. The border of southwest China is mainly mountainous, with diverse climatic changes, so it is necessary to select a Bt maize suitable for the local environment. The field trial results of our study indicated that the DBN3601T maize is highly resistant against oriental armyworm in this area. Compared with the chemical pesticide control of oriental armyworms, the application of Bt maize is more friendly to the environment [35]. However, from the experience of growing Bt maize in other countries, with the long-term, large-scale cultivation of Bt maize, the development of resistance by target pests is a major concern [36]. Until now, there have been multiple reports indicating the development of resistance in certain target pests to some Bt maize types, such as the *Busseola fusca* (Fuller) to the Cry1Ab insecticidal protein [37], fall armyworm to the Cry1Ab and Cry1F insecticidal proteins [38,39], and the *H. zea* to the Cry1Ab and Cry1A.105 insecticidal proteins [40,41].

Managing the resistance of Bt maize to target pests is key to the large-scale sustainable cultivation of transgenic maize [42]. Currently, resistance management is mainly conducted using the high-dose/refuge strategy [43,44]. The high-dose strategy refers to the expression of Bt maize insecticidal proteins at 25 times more than the 99% lethal dose of sensitive target pests, while at the same time planting conventional maize around the Bt maize to provide a certain amount of living space for the sensitive individuals of the target pests. The commonly used strategies include structural refuge (planting conventional maize around Bt maize at a certain ratio), seed mixture refuge (mixing Bt maize and conventional maize seeds in a certain proportion before planting), and natural refuge (planting conventional maize near Bt maize) [45]. While this strategy is scientifically sound, the strict implementation of this strategy is the key to resistance management. For example, in Brazil, during the early stages of planting Bt maize, the growers and government did not prioritize these pests’ Bt resistance management strategies. This led to resistance development in fall armyworms against Bt maize expressing Cry1F (event TC1507) [46]. The quick formulation and implementation of safety management plans to plant Bt crops can better delay the development of resistance in target pests. In the United States and Spain, the cultivation of Bt maize (MON810) with a high-dose expression of Cry1Ab protein and a refuge planting strategy have successfully delayed the development of resistance in *O. nubilalis* (Hübner) and *Diatraea grandiosella* against the MON810 event maize [47,48]. In addition, it has been proposed that the development of resistance can be delayed by regularly updating information about insect-resistant proteins, multigene synergism, alternating Bt maize rotations, integrated control of target pests by chemical pesticides and Bt maize, and Bt maize cultivation only in specific areas [49,50,51]. However, looking at the experience of managing Bt crops over the past 25 years around the globe [52], the resistance management of target pests to Bt crops has not been achieved through the simple application of a single method. Instead, it requires a comprehensive system of management strategies to effectively oversee the cultivation of Bt maize. The strict implementation of these strategies along with resistance monitoring is also crucial [53]. In addition, a high-quality chromosome-level genome of the oriental armyworm was recently assembled via Illumina, PacBio HiFi long sequencing, and Hi-C scaffolding technologies, providing an important genetic basis for developing the management strategy for oriental armyworm in connection with its biological traits such as pesticide resistance [54].

Our research indicates that cultivating Bt-(Cry1Ab+Vip3Aa) maize (event DBN3601T) in southwest China could effectively mitigate insect damage. However, it is necessary to develop an integrated pest management approach by combining Bt maize, entomopathogenic nematodes (EPNs), and other measures for long-term sustainable control of the pest [55,56].

## 4. Conclusions

Several Bt maize events have already obtained safety certificates for commercialization in China since 2019. This study concluded that the DBN3601T event maize could efficiently control the occurrence and damage of the oriental armyworm in southwest China. However, it is necessary to uphold a strategy for managing resistance development of the pest for sustainable application of Bt maize in China.

## 5. Materials and Methods

### 5.1. Maize for Testing

The maize seeds used in the experiment were supplied by Beijing DaBeiNong Biotechnology Co., Ltd. in China (DBN). Bt-(Cry1Ab+Vip3Aa) maize corresponds to the event DBN3601T, and “Huaxingdan88” is the corresponding recipient conventional maize; Bt-Cry1Ab maize corresponds to the event DBN9936, and “Qiushuoyu6” is the corresponding recipient conventional maize; Bt-Vip3Aa maize corresponds to the event DBN9501, and “Zhengdan958” is the corresponding recipient conventional maize. All Bt maize events were planted in Jiangcheng county, Puer, Yunnan province (22°53′33″ N; 101°49′52″ E).

### 5.2. Oriental Armyworm Culture

The oriental armyworm larvae were collected from fields in the Simao district, Puer, Yunnan province, China (22°31′36″ N; 101°29′42″ E), and were continuously reared indoors for experimental purposes. The larvae were reared on an artificial diet [57]. After the larvae were reared indoors until the end of the sixth stage, the larvae were transferred to vermiculite with 30–50% humidity to pupate, and the adult moths were placed in a rearing cage (80 mesh) measuring 50 cm × 50 cm × 50 cm and containing 10% honey solution and white polypropylene rope for oviposition. The larvae and adults were reared in an insect-rearing chamber with a temperature of 25 ± 1 °C, relative humidity of 70 ± 5%, and a photoperiod of 14 h:10 h (L:D).

### 5.3. Tissue Bioassays and Determination of Insecticidal Protein Expression of Bt Maize

Bioassays were conducted on three tissues for each DBN3601T, DBN9936, and DBN9501 maize event, as well as conventional maize. The maize tissues included V5 leaves, R1 silks (unpollinated, 5 cm long), and R2 kernels at various growth stages based on the codes in Abendroth et al. [58], which were used to assess the oriental armyworms. The collected tissues were aliquoted into 24-well culture plates (diameter 2 cm/well), with each well containing 3 g. Subsequently, the first-stage larvae were introduced into each well. The wells were sealed with parafilm, and small holes were punctured using dissecting forceps for ventilation. The maize tissues were replaced every two days. The third- and fifth-stage larvae were individually reared in plastic containers with holes, each containing 30 mL. In each container, 5 g of fresh maize tissue was added, and the tissue was changed daily. The experimental conditions were consistent with those described in Section 2.2. Daily observations were made to determine the survival status of the larvae until the larvae turned to pupae, and the number of larval deaths under different maize tissue treatments were recorded. On the 4th day, the corrected mortality rate of the larvae was calculated. If the larvae showed no response when gently brushed repeatedly with a soft brush, they were considered dead. Each larval stage, Bt maize variety, and maize tissue constituted a treatment, and each treatment was repeated three times. Conventional maize was included as the control.

Insecticidal protein expression of DBN3601T maize tissue (V5-leaf, R1-silk, and R2-kernel) was determined using the sandwich enzyme-linked immunosorbent assay (ELISA); the Cry1Ab/Cry1Ac Quantiplate Kit (Envirologix, Portland, ME, USA) was used to detect Cry1Ab protein and the Vip3A Quantiplate Kit (YouLong Biotech, Shanghai, China) was used for Vip3Aa protein according to the manufacturer’s instructions. The sample processing and data analysis methods are referred to in Wang et al. (2022) [16].

### 5.4. Field Cage Bioassay

The experiment involved a randomized complete block design for both the DBN3601T event and conventional maize “Huaxingdan88”. After sowing, every plot was covered with a mesh cage. The size of the cage was 80 mesh, and the area of each plot was 5 m × 5 m. The above-mentioned maize was planted with a row spacing of 70 cm and plant spacing of 35 cm, and 50 maize plants with the same growth potential were retained at stage V5. Subsequently, 30 first-stage oriental armyworm larvae were placed inside the heart leaf of each maize plant. If there was inclement weather, such as rain or strong winds, within 3 days of infestation, another round of larvae introduction was required. The field inspections began 4 days after larval artificial infestation, which involved confirming the larval population density (larvae per 100 plants), leaf damage score and plant damage incidence. The leaf damage score was determined by following the standards outlined in the Ministry of Chinese Agriculture Bulletin [59]. Inspections were repeated three times for each type of maize with a total of 6 plots.

### 5.5. Control Efficiency Assessment against Naturally Occurring Oriental Armyworm

The natural population dynamics of oriental armyworms were investigated using a method of sampling at five randomly selected points. Ten fields each of Bt-(Cry1Ab+Vip3Aa) (event DBN3601T) maize without insecticide application and conventional maize without insecticide application were selected in Puer, Yunnan province. The first survey was conducted 16 days after the maize was sown and then every 7 days until the maize was R5 stage. A total of 20 fields were surveyed, with 5 points in each field, and the number of oriental armyworm larvae on 20 maize plants was surveyed at each point to calculate the larval density (number of larvae per 100 plants) in each field.

### 5.6. Data Processing and Analysis

Based on the experimental data, the corrected mortality rate, survival rate and control efficacy were calculated using Equations (1)–(4).
Mortality rate (%) = (number of dead test insects after treatment)/(number of test insects supplied before treatment) × 100(1)
Correctional mortality (%) = [(Mortality rate in the treatment group − Mortality rate in the control group)/(1 − Mortality rate in the control group)] × 100(2)
Survival days = The cumulative survival days of all samples/The number of all samples(3)
Control efficacy (%) = [(Larval density of conventional maize − Larval density of Bt maize)/Larval density of conventional maize] × 100(4)

However, a few larvae who were fed the silk tissue of Bt maize could slowly develop into pupal stage, and we take them as the surviving individuals to calculate the mortality rate.

Generalized linear models (GLMs) and Wald’s tests were used to analyze the corrected mortality of different larval stages on different tissues from different maize events and the survival days of different larval stages on different tissues from DBN3601T. Data from field trials were analyzed using the Mann–Whitney U test for significant differences in the larval density of oriental armyworm, leaf damage score, and plant damage incidence between the DBN3601T event maize and conventional maize for the same infection days. The control efficacy of Section 5.4 was analyzed using a one-way ANOVA with Ducan’s multiple comparisons to determine significant differences in different infection days of oriental armyworm. In Section 5.5, the control efficacy in different maize growth stages was analyzed using the Kruskal–Wallis nonparametric test. All significance analyses were performed using SPSS24.0 (IBM, Armonk, NY, USA) software.

## Figures and Tables

**Figure 1 toxins-16-00134-f001:**
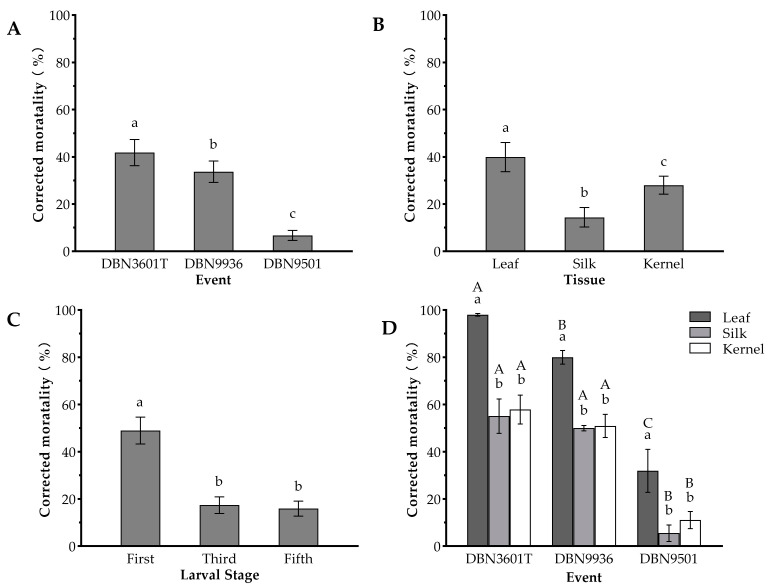
Average-corrected mortality of oriental armyworm larvae 4 days after infestation across all Bt maize events, tissues or larval stages. (**A**) Corrected mortality with each event averaged over tissues and larval stages. (**B**) Corrected mortality with each tissue averaged over events and larval stages. (**C**) Corrected mortality with each larval stage averaged over events and tissues. (**D**) Control efficiency of the different tissues of different events of the first-stage larvae. (**A**–**C**) Different lowercase letters above the error bars indicate significant differences after Wald’s test (*p* < 0.05). (**D**) Different lowercase letters above the error bars indicate significant differences among the corrected mortality of first-stage larvae infesting the different tissues at the same event, and different capital letters above the error bars indicate significant differences among the corrected mortality of first-stage larvae infesting the same tissue at different events (Wald’s test, *p* < 0.05).

**Figure 2 toxins-16-00134-f002:**
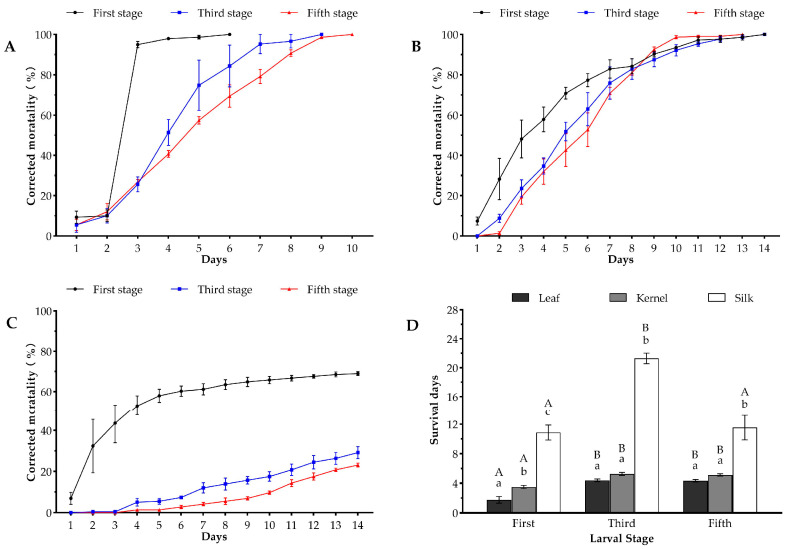
Corrected mortality line graph and survival days of oriental armyworm larvae at different stages infesting various tissues of DBN3601T maize. (**A**) Leaf tissue. (**B**) Kernel tissue. (**C**) Silk tissue. (**D**) Mean survival days at the larval stage. Different lowercase letters above the error bars indicate a significant difference in the survival days of the same-stage larvae feeding on different tissues, and different capital letters above the error bars indicate a significant difference in the survival days of different-stage larvae feeding on the same tissue (Wald’s test, *p* < 0.05).

**Figure 3 toxins-16-00134-f003:**
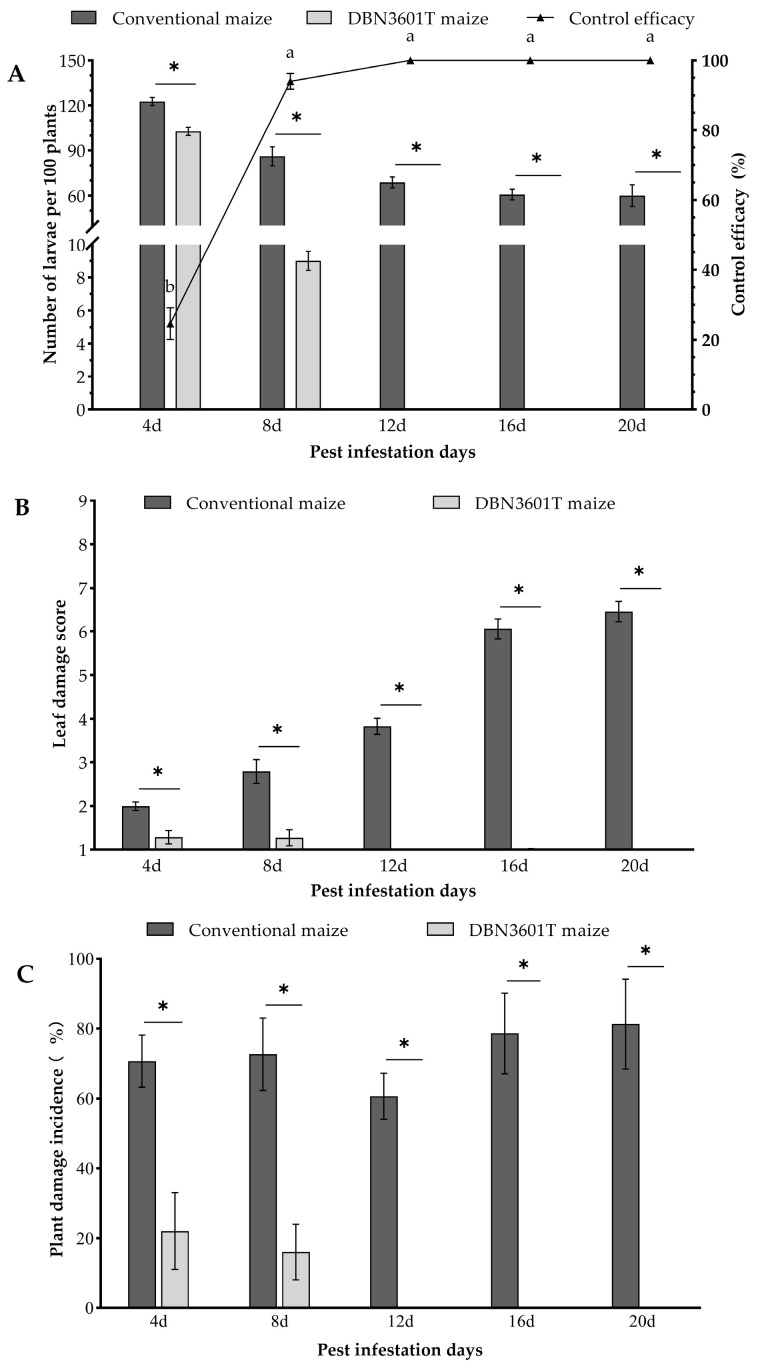
The control effects of DBN3601T against oriental armyworm. (**A**) Larval population density and control efficacy of the DBN3601T maize. (**B**)**.** Leaf damage score. (**C**) Plant damage incidence. The asterisk represents a significant difference in the number of larvae per 100 plants, leaf damage score or plant damage incidence between conventional maize and DBN3601T maize on the same infestation days (Mann–Whitney U test, *p* < 0.05). Different lowercase letters above the error bars indicate a significant difference in the control efficacy of the DBN3601T maize on different infestation days (Duncan’s test, *p* < 0.05).

**Figure 4 toxins-16-00134-f004:**
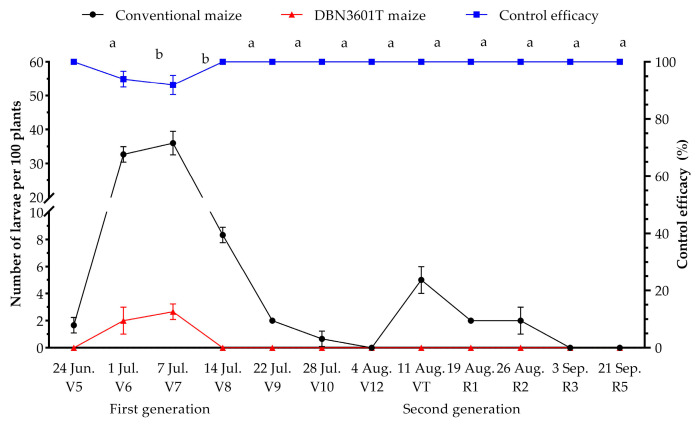
Natural population dynamics of oriental armyworm in Puer, Yunnan province in 2023 and the control efficacy of DBN3601T maize. Different lowercase letters indicate a significant difference in control efficacy at different maize growth stages (Kruskal–Wallis nonparametric test, *p* < 0.05).

**Table 1 toxins-16-00134-t001:** Bt insecticidal protein contents of various tissues expressed by DBN3601T maize plants.

Growth Stage	Tissue	Cry1Ab (μg/g)	Vip3Aa (μg/g)	Total Bt Protein (μg/g)
V5	Leaf	64.68 ± 1.09	5.01 ± 0.09	69.69 ± 1.18 a
R1	Silk	5.83 ± 0.30	1.50 ± 0.01	7.32 ± 0.31 b
R2	Kernel	9.85 ± 0.70	1.84 ± 0.05	11.69 ± 0.75 c

The data in the table are the averages ± standard errors. μg/g: micrograms of Bt protein per gram of lyophilized powder. Different lowercase letters indicate a significant difference in Bt protein content between tissues (Duncan’s test, *p* < 0.05).

## Data Availability

No new data were created or analyzed in this study. Data sharing is not applicable to this article.

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
