# Peer review of "Insecticidal Effects of Transgenic Maize Bt-Cry1Ab, Bt-Vip3Aa, and Bt-Cry1Ab+Vip3Aa against the Oriental Armyworm, Mythimna separata (Walker) in Southwest China"

_toxins, 2024, doi:10.3390/toxins16030134_

Round 1
Reviewer 1 Report
Comments and Suggestions for Authors
This manuscript demonstrated insecticidal activity of transgenic maize expressing Cry1Ab and Vip3Aa against the oriental armyworm. The result was not unexpected since similar results have been reported for many transgenic plants expressing Bt Cry & Vip3A toxins. The most interesting result is the field trial that gave very high control efficacy (92-100%). It would be good if additional data is added, and more explanation is required as follow;
1. Protein production level (Cry1Ab & Vip3Aa) from different transgenic maize events (DBN3601T, DBN9936, DBN9501) should be quantified. The amount of the proteins expressed in different tissues (leaf, silk, kernel) is also important and should be determined. The amount of the proteins produced in each plant as well as in each tissue would give a meaningful interpretation of figure 1 and figure 2.
2. Different stages of insect larvae (1st, 3rd, 5th-instars) were used in some experiments (figures 2 & 3). Since the experiment took up to 20 days, some insects especialy the 5th-insars might turn to pupa. This notion should be clarified in the revised manuscript.
3. Captions for figure 2 should be revised (Line 132). (B) Silk tissue? (C) Kernel? Both did not go along with (D).
4. Keep concistency when writing insect names. Either common name or scientific name can be used but should not mix up. For example, Line 188 ".....fall armyworm, Helicoverpa zea,...." should change to "...Spodoptera frugiperda, Heliceverpa zea..." or fall armyworm, corn earworm...".
5. The last paragraph of the discussion part is not necessary (Lines 274-294).
Comments on the Quality of English Languagenone
Author Response
1: Protein production level (Cry1Ab & Vip3Aa) from different transgenic maize events (DBN3601T, DBN9936, DBN9501) should be quantified. The amount of the proteins expressed in different tissues (leaf, silk, kernel) is also important and should be determined. The amount of the proteins produced in each plant as well as in each tissue would give a meaningful interpretation of figure 1 and figure 2.
Response: Accepted. We have added the determination data of Bt proteins expressed in DBN3601T maize (DBN9936 × DBN9501) in Table 1 (Lines 146-149) as well as the sections of the results, discussion and method.
2: Different stages of insect larvae (1st, 3rd, 5th-instars) were used in some experiments (figures 2 & 3). Since the experiment took up to 20 days, some insects especially the 5th-insars might turn to pupa. This notion should be clarified in the revised manuscript.
Response: Revised (lines 405-406). A few larvae who fed the silk tissue of Bt maize could slowly develop into pupal stage, and we take them as survival individuals to calculate mortality rate.
3: Captions for figure 2 should be revised (Line 132). (B) Silk tissue? (C) Kernel? Both did not go along with (D).
Response: Revised. The title of Figure 2 is a typo. (B) represents the mortality curve after feeding on kernel tissue, and the first, third, and fifth instar larvae all reached 100% mortality, while (C) represents the mortality curve after feeding on silk tissue, and the first, third, and fifth instar larvae did not reach 100% mortality. The survival days in (D) are the average survival days from the larvae eating the tissue to turn pupae or die during the larval phase. All data are in the larval stage. We have corrected “(B) Silk tissue” with “(B) Kernel tissue”, “(C) Kernel tissue” with “(C) Silk tissue”, and “(D) Survival days at larval stage” with “Mean survival days at larval stage” (Lines 138-141).
4: Keep concistency when writing insect names. Either common name or scientific name can be used but should not mix up. For example, Line 188 ".....fall armyworm, Helicoverpa zea,...." should change to "...Spodoptera frugiperda, Heliceverpa zea..." or fall armyworm, corn earworm...".
Response: Revised. We have replaced “fall armyworm” with “S. frugiperda” (Line 200) as well as others.
5: The last paragraph of the discussion part is not necessary (Lines 274-294).
Response: We really appreciate your valuable suggestion. We have deleted the final paragraph and revised the discussion section (Lines 295-299).
Reviewer 2 Report
Comments and Suggestions for Authors
This is a well-written, scientifically sound manuscript that reports the value of a specific Bt event (DBN3601T) for management of the oriental armyworm. The methods are well documented, and the results (including figures are informative and convincing). Given the quality of the writing, the presentation and discussion of data, I think the manuscript requires little revision to be suitable for publication.
I have no criticisms or suggestions regarding content and my only suggestion for revision are some minor changes to language in the manuscript. I made some notations in the pdf and had two broader points for change. First, the manuscript varies between using and not using an Oxford comma (a, b, and c); I think it should have a comma before the and throughout. Second, the authors use the phrase "instar larvae" but my understanding (and the Oxford English dictionary) is that "instar" should only be used as a noun, not and adjective. So I suggest changing these to "stage larvae" throughout.

Comments on the Quality of English LanguageJust notes as above.
Author Response
First, the manuscript varies between using and not using an Oxford comma (a, b, and c); I think it should have a comma before the and throughout. Second, the authors use the phrase “instar larvae” but my understanding (and the Oxford English dictionary) is that “instar” should only be used as a noun, not and adjective. So I suggest changing these to “stage larvae” throughout.
Response: Revised. We checked about 21 places in the full text and corrected all "a, b and c" to" a, b, and c". We replaced all "instar larvae" with "stage larvae" in the full text. We also accept all the modifications in the attachment and make corrections, such as replace “their” with “pesticide” (Lines 52), delete “the” (Line 10), and add “therefore” (Line 52) etc.
Reviewer 3 Report
Comments and Suggestions for Authors
The Manuscript ID: toxins-2871895 examines insecticidal effects of transgenic maize Bt-Cry1Ab, Bt-Vip3Aa and Bt-Cry1Ab+Vip3Aa against the oriental armyworm, Mythimna separata (Walker) in Southwest China. It addresses an important and interesting subject. As the current and sound trends shift towards sustainable agriculture with minimal chemical pesticide use, planting Bt-(Cry1Ab+Vip3Aa) (event DBN3601T) maize in southwest China can efficiently prevent and control the occurrence of oriental armyworm damage. The results also indicated that the insecticidal effect of various Bt maize tissues took an order in leaf > kernel > silk. The authors also referred to and discussed integrated strategies that are necessary for managing resistance development of the pest for sustainable application of Bt maize in China. Generally, the subject is worth publication and the authors did a good job but further insights should have improved this research. I’d suggest the following points:
1) A high-quality chromosome-level genome of the studied oriental armyworm was recently assembled via Illumina, PacBio HiFi long sequencing, and Hi-C scaffolding technologies (Xu et al. 2023). This genome assembly should be stated in the article as it can offer a significant genetic resource for contributing in the development of its management strategies.
REF: Xu, C., Ji, J., Zhu, X. et al. Chromosome level genome assembly of oriental armyworm Mythimna separata. Sci Data 10, 597 (2023). https://doi.org/10.1038/s41597-023-02506-3
2) Following the same line of thinking that contributes in the development of integrated management strategies against the oriental armyworm, some basic data are showing high susceptibility of oriental armyworm to certain entomopathogenic nematode (EPN) species and promise for effective control (e.g., Patil et al. 2020).
REF: Patil, J.; Vijayakumar, R.; Linga, V.; Sivakumar, G. Susceptibility of Oriental armyworm, Mythimna separate (Lepidoptera: Noctuidae) larvae and pupae to native entomopathogenic nematodes. J. Appl. Entomol. 2020, 1–8.
3) These EPNs can kill and recycle in their Mythimna separata host, which bodes well for EPNs’ exploitation in long-term and safe pest management. So, certain approaches suggested to optimize EPN genetics and applications for the integrated management of insect pests should be followed (Abd-Elgawad 2023).
REF: Abd-Elgawad, M.M.M. Optimizing entomopathogenic nematode genetics and applications for the integrated management of horticultural pests. Horticulturae 2023, 9, 865. https://doi.org/10.3390/horticulturae9080865
4) A few typos was found in the MS and should be corrected (e.g., in specific areas [52-54] However, looking).
At least, the three above-mentioned references should be included and discussed.
Therefore, I would suggest resubmitting after major revision.
Comments on the Quality of English Language
Minor editing of English language required
Author Response
- A high-quality chromosome-level genome of the studied oriental armyworm was recently assembled via Illumina, PacBio HiFi long sequencing, and Hi-C scaffolding technologies (Xu et al. 2023). This genome assembly should be stated in the article as it can offer a significant genetic resource for contributing in the development of its management strategies.
REF: Xu, C., Ji, J., Zhu, X. et al. Chromosome level genome assembly of oriental armyworm Mythimna separata. Sci Data 10, 597 (2023). https://doi.org/10.1038/s41597-023-02506-3
Response: We accepted your suggestion and cited the literature in the discussion section (Lines 289-294).
- Following the same line of thinking that contributes in the development of integrated management strategies against the oriental armyworm, some basic data are showing high susceptibility of oriental armyworm to certain entomopathogenic nematode (EPN) species and promise for effective control (e.g., Patil et al. 2020).
REF: Patil, J.; Vijayakumar, R.; Linga, V.; Sivakumar, G. Susceptibility of Oriental armyworm, Mythimna separate (Lepidoptera: Noctuidae) larvae and pupae to native entomopathogenic nematodes. J. Appl. Entomol. 2020, 1–8.
Response: Accepted. We added this part and cited the literature in the discussion (Lines 295-299).
- These EPNs can kill and recycle in their Mythimna separata host, which bodes well for EPNs’ exploitation in long-term and safe pest management. So, certain approaches suggested to optimize EPN genetics and applications for the integrated management of insect pests should be followed (Abd-Elgawad 2023).
REF: Abd-Elgawad, M.M.M. Optimizing entomopathogenic nematode genetics and applications for the integrated management of horticultural pests. Horticulturae 2023, 9, 865. https://doi.org/10.3390/horticulturae9080865
Response: Accepted. We cited and stated the literature in in the discussion (Lines 295-299).
4) A few typos was found in the MS and should be corrected (e.g., in specific areas [52-54] However, looking).
Response: Revised. It is our typos, we have corrected all (Line 284).
Round 2
Reviewer 1 Report
Comments and Suggestions for Authors
Insecticidal protein production level in different Bt-transgenic maize events is quite significant and could be the main reason for differences in insecticidal activity of those transgenic maize events. This data should be mentioned at the beginning of Result section (section 2.1) as well as in abstract, discussion and conclusion.
Comments on the Quality of English Languagenone
Author Response
Insecticidal protein production level in different Bt-transgenic maize events is quite significant and could be the main reason for differences in insecticidal activity of those transgenic maize events. This data should be mentioned at the beginning of Result section (section 2.1) as well as in abstract, discussion and conclusion.
Response: Accepted. We presented the Bt protein data in the Abstract section (Lines 14-16), Results section 2.1 (Lines 84-88) and Discussion section (Lines 211-213).
Reviewer 3 Report
Comments and Suggestions for Authors
Accepted
Comments on the Quality of English LanguageMinor editing of English language required
Author Response
Accepted
Response: Thank you for your comment.